# Forecasting Patient Early Readmission from Irish Hospital Discharge Records Using Conventional Machine Learning Models

**DOI:** 10.3390/diagnostics14212405

**Published:** 2024-10-29

**Authors:** Minh-Khoi Pham, Tai Tan Mai, Martin Crane, Malick Ebiele, Rob Brennan, Marie E. Ward, Una Geary, Nick McDonald, Marija Bezbradica

**Affiliations:** 1ADAPT Centre, D02 PN40 Dublin, Ireland; tai.tanmai@dcu.ie (T.T.M.); martin.crane@dcu.ie (M.C.); malick.ebiele@adaptcentre.ie (M.E.); rob.brennan@ucd.ie (R.B.); marija.bezbradica@dcu.ie (M.B.); 2School of Computing, Dublin City University, D09 Y074 Dublin, Ireland; 3School of Computer Science, University College Dublin, D04 V1W8 Dublin, Ireland; 4St James’s Hospital, D08 NHY1 Dublin, Ireland; maward@stjames.ie (M.E.W.); ugeary@stjames.ie (U.G.); 5School of Psychology, Trinity College Dublin, D02 F6N2 Dublin, Ireland; nmcdonld@tcd.ie

**Keywords:** electronic patient records, multimodal deep learning, explainable AI

## Abstract

Background/Objectives: Predicting patient readmission is an important task for healthcare risk management, as it can help prevent adverse events, reduce costs, and improve patient outcomes. In this paper, we compare various conventional machine learning models and deep learning models on a multimodal dataset of electronic discharge records from an Irish acute hospital. Methods: We evaluate the effectiveness of several widely used machine learning models that leverage patient demographics, historical hospitalization records, and clinical diagnosis codes to forecast future clinical risks. Our work focuses on addressing two key challenges in the medical fields, data imbalance and the variety of data types, in order to boost the performance of machine learning algorithms. Furthermore, we also employ SHapley Additive Explanations (SHAP) value visualization to interpret the model predictions and identify both the key data features and disease codes associated with readmission risks, identifying a specific set of diagnosis codes that are significant predictors of readmission within 30 days. Results: Through extensive benchmarking and the application of a variety of feature engineering techniques, we successfully improved the area under the curve (AUROC) score from 0.628 to 0.7 across our models on the test dataset. We also revealed that specific diagnoses, including cancer, COPD, and certain social factors, are significant predictors of 30-day readmission risk. Conversely, bacterial carrier status appeared to have minimal impact due to lower case frequencies. Conclusions: Our study demonstrates how we effectively utilize routinely collected hospital data to forecast patient readmission through the use of conventional machine learning while applying explainable AI techniques to explore the correlation between data features and patient readmission rate.

## 1. Introduction

Effective risk management in healthcare is crucial as it safeguards patient safety, optimizes resource utilization, and ensures the overall efficiency of healthcare systems, thereby mitigating potential adverse events and enhancing the quality of patient care [1]. Healthcare-associated infections are an example of a key risk for modern healthcare. To be able to assess and control this risk, it is essential for hospitals to have reliable mechanisms to anticipate future events, such as in-hospital infection outbreaks or transmission. As the digital transformation of healthcare becomes more mature, there is a need for new ways to leverage the available healthcare data for the prediction of infection, treatment outcomes, and risk management. The readmission rate is recognized as one of the pivotal metrics for assessing the effectiveness of patient treatment [2]. Clancy et al. [2] define hospital readmission as occurring when a patient returns to the hospital after initial discharge within 30 days. By forecasting individual patients’ readmission risks, interventions can be identified to reduce costs and propose innovative strategies for preventing readmissions [3]. This motivation drives our research effort to evaluate the suitability of typical healthcare data to forecast 30-day readmission rates within one of the national acute hospitals in Ireland as part of a larger research project to improve the prevention and control of healthcare-associated healthcare infections (PCHCAI).

This particular interdisciplinary PCHCAI project, ARK-Virus, has brought together researchers in human factors, health systems, data governance, data analytics, infection control, and microbiology [4]. The ARK-Virus project took a systems approach to managing the risk of healthcare-acquired infection in an acute hospital setting, supported by the ARK (Access Risk Knowledge) Platform. This approach uses the Cube socio-technical systems analysis methodology, which requires evidence-based analysis, like the predictive machine learning (ML) data and deep learning (DL) analysis methods described in this paper. The analysis techniques must be suitable for the data available and the deployment context of risk management.

Conventional machine learning (ML) models, characterized by interpretability, ease of implementation, and well-established frameworks [5], offer practical advantages in forecasting hospital readmission, providing informative insights into risk factors and facilitating trust among healthcare professionals. Their simplicity, limited data requirements, and alignment with clinical reasoning make them effective tools for enhancing risk management in healthcare.

This research explores the potential of the different conventional ML models on this dataset to predict patient outcomes. We also incorporate several well-known DL models into our experiments for comparison. Additionally, we emphasize model explainability, aiming to identify specific data attributes that are likely to impact patient readmission, since explainability is one of the most important requirements in healthcare [6].

In summary, this paper addresses two key research questions:How effectively can conventional ML models utilize diverse data types within Irish healthcare discharge records to predict patient outcomes, such as the 30-day readmission rate?How can we generate clinically meaningful explanations for these predictive models using a popular explainable AI technique?

Our main contributions are as follows:We evaluate various conventional ML algorithms, as well as a set of DL methods, for predicting patient readmissions using routinely collected discharge records data from an Irish acute hospital that encompass patient demographics, their clinical information, diagnosis codes, and previous hospitalization.We extensively assess the importance of each data feature in predicting patient readmission by analyzing the impact of individual group features on the classification score, while also testing with different data sampling methods to address the class imbalance.To gain deeper insights into the models’ predictions, we use SHAP visualizations to examine how patient clinical information and diagnosis codes are related to 30-day readmissions. Finally, we interpret the significance of these visualizations.

Overall, the structure of our paper is as follows: We present an overview of prior research in Section 2. Section 3 provides an introduction to our data sources and the methodology we employ. In Section 4, we present the outcomes of our benchmarking models and accompany these with visualizations depicting the importance of the features. Lastly, in Section 5, we engage in a discussion concerning the results, and limitations, and outline possible avenues for future research.

## 2. Related Work

### 2.1. Multimodal Machine Learning in Patient Readmission Forecast

Various research studies have explored the task of predicting patient readmission with multiple different time frames, such as 30, 180, or 365 days, depending on their focus. In our study, our primary focus lies in predicting readmissions within a 30-day window. The surge in data potentially available for clinical and safety decision making as a result of digital transformation is not only marked by a substantial increase in volume but also by a growing complexity and diversity in their forms. These challenges contribute to a notable rise in the publication of research papers, particularly in the multimodal machine learning topics, since multimodal methods address the data diversity.

Many researchers [7,8,9,10,11] have substantiated the advantages of amalgamating diverse data types extracted from patients’ electronic health records (EHRs). These encompass patient demographics, medical diagnoses, vital signs, lab test results, and historical hospital visit records, all of which contribute to the prognosis of future readmissions. These studies have adeptly incorporated multimodal configurations into a variety of machine learning models to cater to this task, yielding superior performance compared to single-modal methodologies.

Hence, we aim to introduce and address the unique characteristics of our discharge records data, which may differ from data in other research. For example, lab tests and vital signs are pivotal features in both [9,10], whereas they were not available in our study. Additional supplementary information can also be leveraged for readmission prediction, such as integrating clinical prescription data with weather and air quality records [12], or patient-reported outcome measures (short, self-completed questionnaires) [13]. However, the effectiveness of these measures largely depends on their design, implementation, and the timing of data collection, and they are not always readily available. Consequently, in Section 3.1.1, we will provide an introduction to our electronic data, highlighting the associated characteristics, and thereby distinguishing our approach from these aforementioned studies.

Given these differences, we shift our emphasis to the clinical diagnostic aspects, particularly the utilization of International Classification of Diseases (ICD) codes, since these are commonly available to access in the global healthcare system. Many studies have employed diverse techniques to extensively harness the information encapsulated in ICD code names for tasks related to predicting patient outcomes. These approaches span from straightforward tree-based models [10,14] to more advanced deep language models, such as convolutional networks [15] and recurrent networks [7,8,16,17]. Studies have found that instead of relying exclusively on medical code names, leveraging the contextual descriptions of clinical concepts can amplify the effectiveness of predictive models [15,18]. In our experiment, we seek to conduct a comparative analysis between the use of code names and descriptions, in conjunction with other variables, to reassess this claim. Our position asserts that for the conventional model we employ for this dataset, utilizing clinical codes rather than textual descriptions is seen to offer more effectiveness to the models’ accuracy.

It is worth noting, however, that these studies typically do not address the challenge of class imbalance (the imbalance of classes can be defined as when the number of one type of data points outnumbers the instances of another type), which is a prevalent and ubiquitous issue when utilizing many types of electronic data from the healthcare domain for AI applications. While previous studies [7,9] suggest the introduction of additional weights in their cost function to mitigate this problem, they report that this approach is not particularly effective [7]. In contrast, our approach directly tackles this issue by comparing a variety of data sampling techniques, from random sampling to KNN-based sampling, which significantly improved the overall performance metrics of our methods.

Furthermore, the existing research papers to date often lack adequately transparent explanations for their models’ performance. This deficiency is primarily attributed to the fact that such explanations are typically tailored to specific models and cannot be readily applied to others [14], or they necessitate the integration of additional components like attention modules [16,18]. In our research, we have adopted a widely recognized and universally applicable Shapley additive explanations (SHAP) method, regardless of the specific architecture of the machine learning models employed. While SHAP is a popular tool for explaining feature importance in ML models, many studies report SHAP values for only a single model. In practice, different models may capture and represent feature relationships with the target variable in diverse ways, leading to variations in SHAP values across models. Therefore, we not only use SHAP to investigate the relevance of patient information and diagnostic code groups that may influence early readmission but also aggregate SHAP values across multiple models to ensure the stability and reliability of our findings.

### 2.2. Explainable AI in Healthcare Domain

There has been a growing demand for transparency of AI models to be utilized in the healthcare sector [6]. In our research, we advocate utilization of the SHAP method [19], which is based on the Shapley value [20] (we highly recommend looking at this extensive book for a more intuitive explanation of SHAP value https://christophm.github.io/interpretable-ml-book/shapley.html, (accessed on 30 August 2024)), to visualize the importance of model features. SHAP values possess the advantage of being model-agnostic, making them universally applicable in conjunction with any machine learning models. This represents a straightforward approach that offers additional interpretability without affecting the models’ parameters, thus preventing compromising model accuracy.

A recent comprehensive survey [6] provides an extensive list of explainable methods for interpreting model predictions and investigating their underlying behaviors. The survey notes that in many previous studies utilizing EHRs as data, SHAP is often the preferred choice among explainability methods. However, it is worth noting that these studies commonly overlook the inclusion of patients’ diagnosis codes and descriptions [21,22,23]. Thus, in our study, we illustrate how this approach can provide useful information on which factors potentially influence patient readmissions based on their prior clinical diagnosis codes and the descriptive terms in our dataset.

## 3. Method

### 3.1. The Collected Data

#### 3.1.1. Data Description

The data used for this retrospective study comprised an anonymized sample of typical operational Irish healthcare electronic data, collected daily from inpatients at St. James’ Hospital, Dublin, Ireland. As such, they represent the diversity of interlinked data that are needed for this sort of analysis and highlight challenges with data consistency, sparseness, and other quality factors that are often found in operational data. Demonstrating methods that can overcome these challenges is a major goal of this study.

In Irish healthcare settings, inpatient data is digitally collected through the Hospital Inpatient Enquiry (HIPE) system. HIPE plays a pivotal role in Irish healthcare data collection, generating approximately 1.7 million records annually (information for HIPE can be found at https://hpowp.com/hipe-home/, (accessed on 30 August 2024)). A HIPE discharge record is created when a patient is discharged from (or dies in) a hospital. It encompasses patients’ demographic, clinical, and administrative data for each bed day, including discharge codes noted by medical administrators, which reflect patient conditions for a discrete episode of care. An episode of care begins at admission to hospital, as a day or inpatient, and ends at discharge from (or death in) that hospital.

The HIPE data in our study are neatly organized in a table and include both structured and semi-structured fields, covering a time period from 2018 to 2022 and consisting of almost 1 million rows. Each row in this dataset represents one patient’s stay at the hospital, recorded every night. Furthermore, the dataset includes patients’ demographic information as well as their hospitalization, including the episode admission and discharge time, medical wards they were admitted to and discharged from, and the speciality of that ward. The data features are described in Appendix A, and the main statistical description of the dataset is reported in Table 1.

Besides patient demographics such as gender and county of residence, the dataset also includes information on up to 29 diagnoses and 29 medical procedures for each hospital stay. These are entered as clinical codes by hospital discharge administrators based on the free-text medical notes from doctors. These clinical codes are categorized using the widely recognized guidelines within the HIPE system, namely Irish Coding Standards (ICS) [24]. The ICS are guidelines for the collection of data for all discharges from hospitals in Ireland and the 10th Edition ICD-10-AM/ACHI/ACS classification system. The ICD-10 system, published by the World Health Organization (WHO) [25], is a standardized method for categorizing and labeling diagnoses, symptoms, and procedures related to hospital care. Additionally, the dataset contains codes and descriptions for Diagnosis Related Groups (DRG) and Major Diagnostic Categories (MDC). These codes help categorize different diagnoses for the purpose of healthcare reimbursements. In total, the dataset encompasses 9851 distinct diagnosis and procedure codes. Some illustrative examples are presented in Table 2.

#### 3.1.2. Data Cleaning and Feature Engineering

In this section, we present a comprehensive overview of our data processing pipeline. We begin by selectively retaining episodes that occurred between 1 January 2018 and 28 February 2022, stripping episodes that do not have a final discharge date or ones that are missing an admission date as they would likely generate noises in our data. In order to maintain data consistency, we exclude records associated with patients who were either deceased during hospitalization or were transferred to another healthcare facility. These cases are considered outliers since our primary objective is predicting patient readmission within 30 days, and these situations may not align with that goal [10].

For numerical data features, we apply standardization to ensure numerical stability, while categorical data features are converted into one-hot encoding matrices. These preprocessing steps are not necessary for LightGBM and CatBoost, as these frameworks handle them internally.

The sequential data from patients’ past hospital visits hold valuable insights for predicting their future hospitalizations, a concept well-supported by prior research [7,26,27]. Thus, we have manually integrated cumulative features based on the patients’ historical admissions. The representative features for each data sample include cumulative metrics from the past t episodes, such as the *total length of stay since the last t episodes*, *the number of ICU admissions in the last t episodes*, and *the counts of diagnoses and procedures from prior admissions within the last t episodes*. We also present an ablation study in Figure A3 to examine the impact of t, which appears to significantly influence model performance.

For the medical codes in our dataset, studies show that utilizing the most granular levels of medical codes may not be imperative for effective predictions [7,8,14,16,17]. Following their footsteps, we follow this approach by grouping the first three digits of ICD-10 diagnosis codes and five digits of ICS procedure codes. This grouping significantly reduces the number of unique codes (as shown in Table 1). These characters provide a high-level grouping of diseases that represents the category of the diagnosis, which provides sufficient information for machine learning models to learn in our study. We also report the ablation study for this setting in Figure A1.

For the code’s descriptions, we apply standard text preprocessing techniques, including converting text to lowercase, and eliminating stopwords, punctuation, and numerical characters. Missing values in these columns are uniformly filled with “None” to ensure consistency. Finally, to maintain data integrity, we remove duplicate rows and empty columns from the dataset. This step ensures that our dataset remains clean and well-structured for analysis.

### 3.2. Models

#### 3.2.1. Data Sampling Method

Our dataset demonstrates a significant class imbalance between the number of readmitted and non-readmitted patients, as shown in Figure 1, with the positive class representing only 12% of the data.

We conducted a power analysis to estimate the required sample sizes for predicting 30-day patient readmission, considering varying effect sizes, as shown in Figure 2. Using a significance level of 0.05 and a desired power of 0.8, our analysis revealed that as the effect size decreases, the required sample size also increases. For smaller effect sizes, such as 0.2, a much larger sample size is necessary to achieve sufficient power, while larger effect sizes, such as 0.8, require fewer samples. Based on our analysis, for an effect size of 0.3, and with the 12% class imbalance ratio taken into account, we estimate that approximately 500–3000 samples for each class would be needed to achieve the desired power.

To tackle this issue, we conduct an extensive experiment to identify the optimal data sampling method, covering the five most commonly used techniques:**Majority Undersampling**: Randomly removes instances from the majority class to balance the class distribution.**Minority Oversampling**: Randomly replicates instances from the minority class to balance the class distribution.**Tomek Link Undersampling** [28]: Identifies and removes majority class samples that form Tomek links (pairs of nearest neighbours from different classes) to clean the decision boundary.**Synthetic Minority Over-sampling Technique (SMOTE)** [29]: Generates synthetic samples for the minority class by interpolating between existing instances.**Adaptive Synthetic Sampling (ADASYN)** [30]: Generates synthetic samples by focusing more on the minority class instances that are near the decision boundary or in regions with more majority class samples.

#### 3.2.2. Term Frequency-Inverse Document Frequency

To encode textual data for ML models, we employ a commonly used technique, namely term frequency-inverse document frequency (TF-IDF) [31]. TF-IDF assesses the importance of individual terms within a text by calculating their frequency. We utilize these encoding techniques for experiments that employ clinical codes or code descriptions as training features for machine learning models. In medical texts, it is common for terms to appear as compound nouns. To try capturing potential associations between these terms, we generate n-gram tokens from the code descriptions. We retain only those terms that appear at least twice in the episode treatment. We also report the ablation study for choosing n in Figure A2. It should be noted that when using the clinical diagnosis codenames rather than descriptions as the training features, these associations are reflected in the codenames due to their hierarchical structure design. Initially, TF-IDF is trained on the vocabulary of the training set and subsequently encodes entire medical paragraphs as single vectors for each episode in the validation set.

#### 3.2.3. Machine Learning Models

In our investigation, we perform a comprehensive benchmarking comparison to identify the most optimal choice among nine models, encompassing both machine learning and deep learning approaches, with the results detailed in Table 3. These methods encompass some of the most well-known conventional machine learning methods: logistic regression, random forest, AdaBoost, XGBoost, CatBoost, and LightGBM. Logistic regression serves as a fundamental algorithm primarily utilized for binary classification tasks, recognized for its simplicity and interpretability [32]. In contrast, random forest operates as an ensemble learning technique, combining multiple decision trees to enhance predictive accuracy while mitigating overfitting risks [33]. AdaBoost, short for adaptive boosting, is tailored to enhance the performance of weak learners by assigning greater weight to misclassified instances [34]. Among the high-performing gradient boosting frameworks, we have selected XGBoost, CatBoost, and LightGBM. Extreme gradient boosting (XGBoost) [35] is recognized for its speed and performance due to features such as regularization, missing value handling, and parallelization, making it particularly suited for structured or tabular data. CatBoost [36] is engineered to manage categorical features natively, eliminating the need for extensive preprocessing like one-hot encoding. LightGBM [37] employs a leaf-wise tree growth algorithm, offering faster training times and improved performance on large datasets. Additionally, we leverage the Optuna tool [38] to optimize hyperparameters across all experimental models, as detailed in Table 3.

#### 3.2.4. Deep Learning Models

We also incorporate representative deep learning models that are frequently utilized for natural language data: CNN, LSTM, and transformer. Convolutional neural networks (CNN), primarily designed for image processing, can also effectively learn from natural language data by employing convolutional filters to capture local feature interactions. In the context of language data, CNNs can automatically identify patterns or dependencies among neighbouring language tokens, thereby facilitating the learning of structured or spatial relationships within the data [40]. Long short-term memory networks (LSTM), a variant of recurrent neural networks (RNN), excel at capturing sequential dependencies in data. When applied to tabular datasets, LSTMs are capable of modeling temporal or sequential relationships between features, particularly when the data have a time series or ordered structure. Transformers [39], characterized by their self-attention mechanism, can effectively manage tabular data by focusing on feature importance across the entire dataset. Unlike CNNs and LSTMs, transformers are effective at capturing complicated feature interactions without being constrained by local or sequential structures, making them well-suited for high-dimensional tabular data with diverse feature relationships. To utilize these models with our feature sets without modifying their architecture, we only use clinical code names and descriptions as training features.

## 4. Results and Discussion

### 4.1. Validation Method and Performance Metrics

Researchers have extensively studied suitable evaluation metrics for classification tasks within the healthcare domain [41]. They have emphasized the importance of not relying solely on a subset of metrics, as doing so could potentially yield misleading results when implementing models in clinical settings. To comprehensively assess model performance, we employ multiple evaluation metrics, including sensitivity, specificity, F1 score and area under the curve score (AUROC). The equations for the first three metrics are provided below:(1)$$\begin{array}{cc} Specificity & =\frac{TN}{TN+FP} \end{array}$$(2)$$\begin{array}{cc} Sensitivity & =\frac{TP}{TP+FN} \end{array}$$(3)$$\begin{array}{cc} F1 & =\frac{2TP}{2TP+FP+FN} \end{array}$$

Here, TP, FP, TN, and FN denote true positives, false positives, true negatives, and false negatives, respectively. Specificity places a greater emphasis on correctly classifying negative samples, while Sensitivity aims to minimize the misclassification of positive instances, making it a critical metric in medical studies [41].

The F1 score represents the harmonic mean of precision and recall, providing a balanced measure that penalizes extreme values of either precision or recall. AUROC, on the other hand, has gained popularity in machine learning [42] due to its favourable properties when dealing with imbalanced classes. It is used to distinguish between positive and negative classes across different threshold settings regardless of class distribution, making it an effective balance measurement. As a result, we have selected AUROC as the balance metric for model comparison.

To assess the robustness of our models, we employ cross-validation techniques, particularly stratified five-fold cross-validation. This approach ensures that each fold maintains reasonable class distribution, mitigating the effects of class imbalance problems when using cross-validation. Following the guidance of [41], we avoid sharing data from the same patients across different folds to prevent introducing bias during the parameter tuning phase. We report the mean and standard deviation of model performance across these folds, and predictions made on the test set are presented for all the experiments.

### 4.2. Model Performance

Table 3 presents a performance comparison among traditional ML and DL approaches, revealing that CatBoost stands out as the leading model in terms of AUROC score, while LightGBM achieves the highest F1 score. It is intriguing that a straightforward algorithm like logistic regression shows a commendable balance in metric scores, likely due to the effectiveness of our robust data engineering pipeline and meticulous hyperparameter tuning. In contrast, the advanced deep learning methods appear to be less suited for this task, as evidenced by their underwhelming AUROC scores. This may be attributed to the fact that these models are solely trained on textual features, potentially overlooking significant predictors and missing out on the advantages offered by feature engineering. Zarghani [43] also found that deeper models such as LSTM tend to overfit on the training set and not be generalizable to the smaller validation subset, resulting in inferior performance compared to methods like XGBoost and LightGBM in predicting patient readmission in diabetic patients. Figure 3 illustrates the ROC curves for these models.

We also conduct a thorough evaluation of the individual features included in our models in order to assess their importance in Table 4. This evaluation involved systematically including each feature one at a time and examining whether there is an improvement in the evaluation score. We perform this setup for all ML models and aggregate the results for each setup. This ensures the consistency of the experiment and proves that the inclusion of newly engineered features is the main contribution to the performance. Otherwise, we include Figure 4 showing the impact of each group of features on the models individually.

As presented in Table 4, it is crucial to note that the inclusion of clinical codes and descriptions has the most significant impact on the performance of the model, leading to an improvement of 3% in the AUROC score. Additionally, our findings agree with the conclusions drawn by Ma et al. [15] regarding the utility of code descriptions. In our case, the inclusion of code descriptions indeed led to performance improvement as opposed to using code names only (Experiments 2 and 3). For more information on this, we attach the detailed evaluation table in Table A1.

Moreover, leveraging data from prior admissions introduces valuable prior knowledge for future decision-making. Experiment 4 received a substantial AUROC score improvement of approximately 3% thanks to the incorporation of cumulative features from previous hospital episodes. As anticipated, we gain significant performance enhancement when all features from various modalities are incorporated. This observation strongly supports the efficacy of multimodal techniques in our predictive model.

It is important to highlight that there is a significant imbalance between specificity and sensitivity in most of our reported experiments in Table 4. Experiments 1–4 exhibit extremely high specificity scores, indicating a strong bias toward the negative class, which constitutes a substantial portion of the dataset. In contrast, Experiment 5, which integrated a sampling technique, achieve a more balanced distribution of scores between specificity and sensitivity, resulting in improved overall balance performance. We also report an ablation study on different choices of data sampling methods in Figure A4.

Overall, traditional models demonstrate a reasonable level of effectiveness in dealing with complex and diverse datasets like patient discharge records when appropriate data processing and engineering techniques are applied.

### 4.3. Feature Importance

In this section, we demonstrate the application of SHAP value visualization to illustrate feature importance for our models. The computation of SHAP values involves systematically perturbing input features and analyzing how these modifications impact the final model predictions. Figure 5 demonstrates the relative rankings of features based on their absolute SHAP value, aggregated across multiple models. In detail, for each feature (see the feature description in Appendix A), we compute its absolute SHAP value, and then we rank the features based on this value. We do this for all ML models we mentioned, and then scale the ranking to 0–1 and average them. The higher the value of average ranking, the more influence it has on patient readmission within the next 30 days. As can be seen, CODENAMES and CODE_DESCRIPTION show a strong influence on patient readmission since they directly reflect the patient’s condition. In contrast, other features display a wide variation in ranking distribution, indicating that different models prioritize different features as important.

In the following figures, the arrangement of features, from top to bottom, signifies their importance, with the most significant features occupying the upper positions. Additionally, each dot on the plot represents an individual data sample, with the colour intensity of each dot reflecting its numeric value when input into the model. The *x*-axis in the visualization displays the SHAP value for each plotted dot. Positive SHAP values indicate that the model is more inclined to classify the corresponding sample as positive (in our context, signifying that the patient is likely to be readmitted within the 30-day window), and negative values suggest the opposite. A SHAP value “0.0” is indicative of minimal impact. Within our focus, we exclusively examine the SHAP values associated with clinical codes and descriptions to assess their contribution to the model’s predictions.

In Figure 6, we focus on what are the top influencing factors that are associated with patient readmission. Firstly, DISCHARGE_DESCRIPTION and DISCHARGE_SPECIALTY emerge as the most crucial feature in our dataset. We hypothesize that these two features reflect the clinical decision at the time of discharge, potentially indicating the guidance for diagnosis, treatment, and destination received by patients upon discharge. Consequently, patients discharged from some specific departments (e.g., ones that often deal with chronic or long-term conditions) might have a higher likelihood of readmission. Additionally, the length of stay (LOS) feature and the total count of previous hospitalizations for a patient carry meaningful predictive values, suggesting that patients with frequent readmission history and longer hospital stays have an increased likelihood of readmission within a month. Lastly, we observed that a higher value in NUM_DIAGNOSIS is associated with an elevated impact on the prediction of patient readmission.

Figure 7 provides an in-depth exploration of the most influential clinical codes on models, as outlined in Table 4. We have selectively chosen to present codes or terms that exhibit clearly distinguishable SHAP value measurements, meaning that their feature value does not vary along the *x*-axis. For those codes or terms that do not meet this criterion, despite having a significant impact, their interpretation is not straightforward in isolation and should be considered in conjunction with other terms or codes. Some of our findings in the feature importance align with medical insights from previous research.

Likewise, ICD-10 code ’Z59’ corresponds to patients facing ’Problems related to housing and economic circumstances’, indicating a link between social factors and readmission rates. Previous studies have substantiated the association between social factors and all-cause unplanned readmissions, highlighting low socioeconomic status and housing instability as prominent contributing factors [44]. We also find ’Z22’ to be one of the minor causes for readmission; it presents a group of codes for ’carrier of infectious diseases’, which is of potential interest to the ARK project. In our interpretation, these clinical codes, indicating the diagnoses of patients’ conditions and their social status, are important predictors of the patients’ 30-day readmissions.

In conclusion, our SHAP value visualization analysis has provided valuable insights into the global observation of how a patient’s diagnosis contributes to the likelihood of their readmission within 30 days. Notably, diseases such as cancer, indicated by clinical codes like ‘96199’ and terms like ‘malignant neoplasm’, emerge as key contributors to increased readmission risks. Furthermore, chronic conditions like ‘J44’ (COPD) and social factors represented by ‘Z59’ are identified as significant determinants. However, being bacteria carriers, as indicated by code ‘Z22’, appears to have minimal impact on the readmission likelihood of patients, likely due to the scarcity of such cases during the study period. As one of our objectives is to study clinical risks, we leave the investigation of the effects of bacteria for our future research.

## 5. Conclusions and Future Works

Patient risk management is crucial for healthcare providers because it ensures patient safety, prevents adverse events, and reduces legal and financial risks. Forecasting patient readmission automatically using AI helps hospitals achieve these goals by allowing timely interventions, better post-discharge planning, and more efficient resource allocation.

In this study, we carry out a performance comparison of multiple conventional ML models on the extensive incorporation of multimodal discharge records data from an Irish acute hospital. Despite the complex and diverse nature of data within the HIPE system, gradient boosting algorithms come out as the top algorithms, demonstrating their strong ability to simultaneously capture patient demographics, historical hospitalization records, and clinical diagnosis codes. Overall, after integrating all these features into the algorithm, we successfully improve the results from a baseline AUROC score of 0.628 to 0.7 averaging over nine models. Although an AUROC score at around 0.7 seems to be moderate, it falls within the typical range observed in an overview study for readmission prediction conducted by Yu and Son [45], which is between 0.51 and 0.93. This also highlights the utility of routinely collected data in hospitals to further improve the outcome of patients in this digital era.

The successful application of the conventional SHAP value technique to dissect the ML models has allowed us to provide visually interpretable figures that illustrate the predictions of the model without compromising accuracy on complex multimodal healthcare data. Notably, through this analysis, we have established a robust correlation between patients’ diagnosis codes and their likelihood of future readmission. This understanding has the potential to empower hospital experts and organizations in anticipating future risks and costs more effectively, ultimately leading to improvements in hospital processes and enhanced patient outcomes. By recognizing that certain common illnesses within Irish hospitals significantly contribute to readmission risks, healthcare providers can tailor interventions and care plans to address the specific needs of patients with these diagnosis codes. This proactive approach has the potential not only to reduce the likelihood of readmissions but also optimize resource allocation, thus fostering a more efficient and patient-centric healthcare system.

Nevertheless, there are several limitations to our study. One challenge when working with clinical codes is their high dimensionality due to their complicated and sparsely populated nature. As evidenced by the approximately 10,000 distinct codes reported in Table 1, sparsity is a significant concern. Conversely, utilizing clinical terms results in a less sparse word embedding matrix, enabling the capture of the hierarchical structure inherent in these clinical codes. Despite our efforts to address this issue through code generalization, it proved to be an ineffective solution—as evidenced in Figure A1—possibly due to loss of information. It is also possible that missing data may contain concealed information that could serve as potential predictors, as has been reported in [46]. We have noticed that other authors have recommended other methods for missing data and their impact on ML classifiers in similar problems [47,48]. In addition to these, we have conducted comprehensive benchmarks to assess different data sampling methods. However, the same configurations have yet to be applied to deep learning models, largely due to the complexities associated with the data. Future research should aim to investigate more sophisticated strategies to effectively address both of these challenges.

In the realm of healthcare, future research may need to delve deeper into the architectures of deep learning models to adapt them to this specific domain for better performance, but it will also require additional efforts to enhance transparency in dealing with their black-box architectures, as well as seeking domain expert’s validation and verification. For instance, language models like CNN [49], transformer [39], or process mining techniques could be explored further to leverage sequential patient visits and improve predictive capabilities while maintaining interpretability.

## Figures and Tables

**Figure 1 diagnostics-14-02405-f001:**
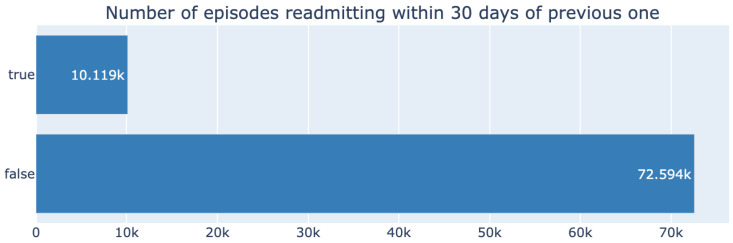
Readmission class distribution in the dataset, which is highly imbalanced toward the negative class.

**Figure 2 diagnostics-14-02405-f002:**
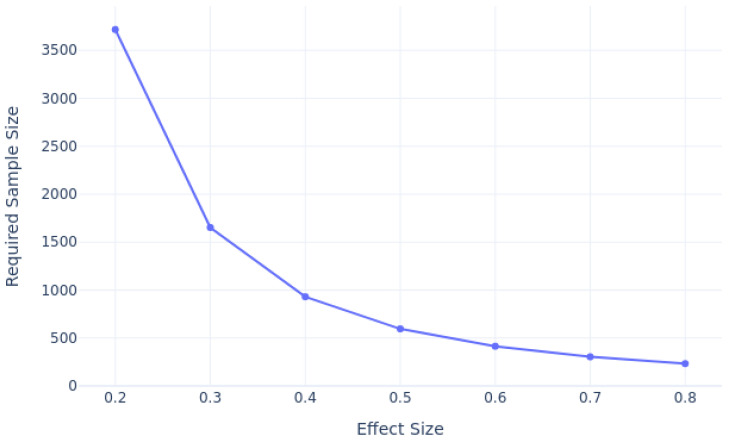
Power analysis with different effect size.

**Figure 3 diagnostics-14-02405-f003:**
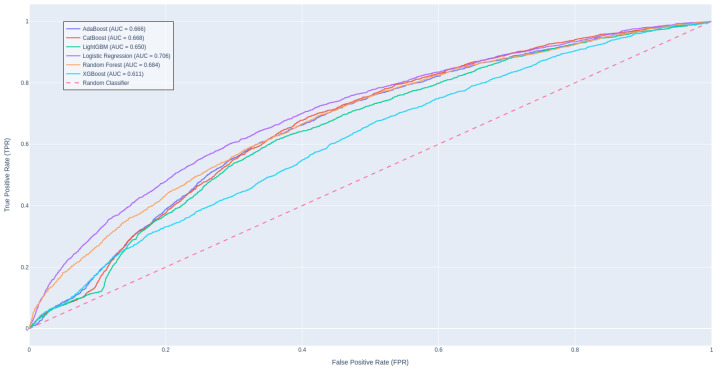
ROC curves for all experiment models.

**Figure 4 diagnostics-14-02405-f004:**
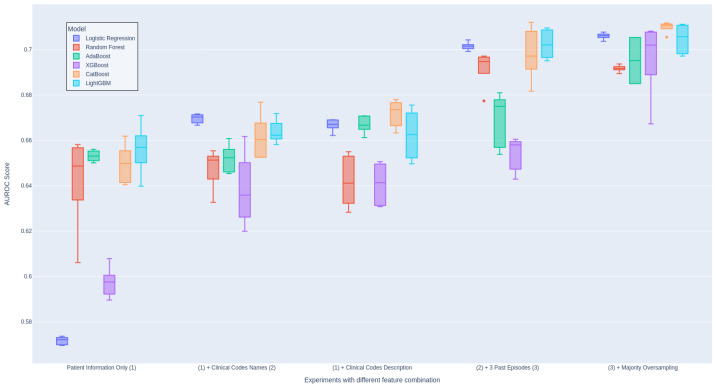
Effectiveness of each data features’ inclusion and techniques applied across different models.

**Figure 5 diagnostics-14-02405-f005:**
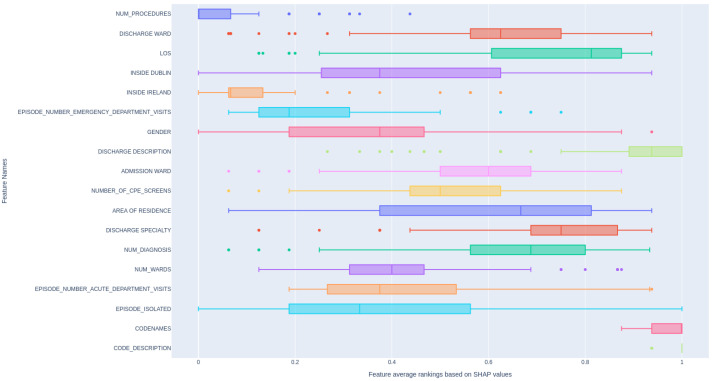
Overall feature ranking based on SHAP absolute values across models.

**Figure 6 diagnostics-14-02405-f006:**
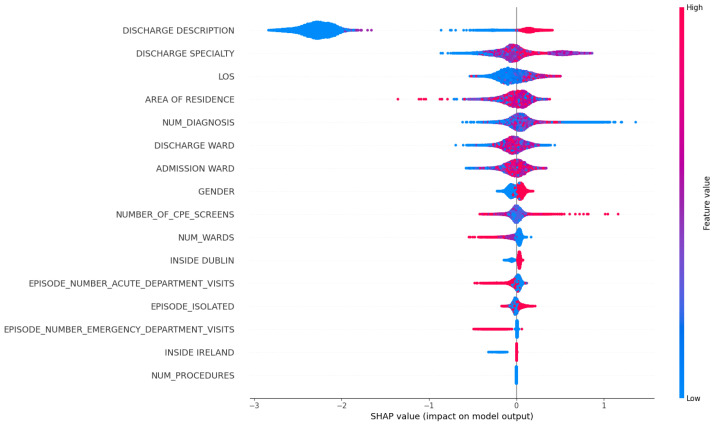
Feature importance based on SHAP values, aggregated and averaged across models.

**Figure 7 diagnostics-14-02405-f007:**
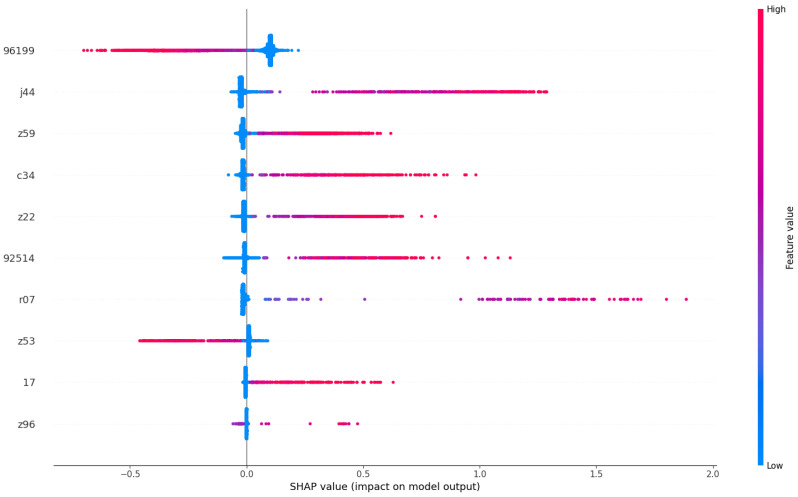
Clinical code importance. We present generalized clinical codes that have a large impact on patient readmission prediction across models.

**Table 1 diagnostics-14-02405-t001:** Overall statistics of the data after processing. Clinical codes include International Classification of Diseases (ICD), Irish Coding Standards (ICS), Diagnosis Related Groups (DRG), and Major Diagnostic Categories (MDC).

Feature Name	Statistics
Number of patients	50,159
Number of episodes	82,713
Highest value of number of clinical visits of a single patient	46
Number of unique clinical codes	9851
Number of unique generalized clinical codes	3771

**Table 2 diagnostics-14-02405-t002:** Example of a data instance where multiple codes might be assigned in an episode. These codes are shown at their most granular level.

Diagnosis Code	Code Description
J44	Chronic obstructive pulmonary
D351	Benign neoplasm of parathyroid gland
I20	Angina
Z22.3	Carrier of other specified bacterial diseases
Procedure Code	Code Description
9555009	Allied health intervention pharmacy
3031500	Subtotal parathyroidectomy
DRG Code	Code Description
K05A	Parathyroid procedures
MDC Code	Code Description
10	Parathyroid procedures

**Table 3 diagnostics-14-02405-t003:** Comparison between conventional ML models and DL models on the test set using their best settings on the validation set found by the Optuna optimizer. Bold colored texts indicate best results of the metrics.

Model	Results on Test Set Using Cross-Validation				
	Balanced Accuracy	Specificity	Sensitivity	F1 Score	AUROC Score
Logistic Regression [32]	**0.646 (0.003)**	0.57 (0.009)	0.72 (0.008)	0.37 (0.003)	0.706 (0.001)
Random Forest [33]	0.633 (0.002)	0.56 (0.01)	0.706 (0.005)	0.364 (0.002)	0.683 (0.013)
AdaBoost [34]	0.63 (0.007)	0.52 (0.0004)	0.75 (0.015)	0.364 (0.006)	0.69 (0.004)
XGBoost [35]	0.64 (0.005)	0.6 (0.009)	0.678 (0.015)	0.37 (0.003)	0.69 (0.009)
CatBoost [36]	0.641 (0.002)	0.51 (0.013)	**0.768 (0.011)**	0.369 (0.002)	**0.71 (0.002)**
LightGBM [37]	0.645 (0.001)	0.61 (0.015)	0.68 (0.016)	**0.378 (0.0013)**	0.704 (0.006)
CNN1D	0.614 (0.01)	0.57 (0.123)	0.655 (0.11)	0.355 (0.009)	0.663 (0.013)
LSTM	0.632 (0.009)	**0.633 (0.052)**	0.63 (0.068)	0.372 (0.008)	0.68 (0.01)
Transformers [39]	0.62 (0.008)	0.566 (0.087)	0.672 (0.078)	0.358 (0.007)	0.67 (0.008)

**Table 4 diagnostics-14-02405-t004:** Comparison of effectiveness between features used in training the ML models. The results are aggregated across ML models and cross-validation folds and benchmarked on our data test set. Bold colored texts indicate best results of the metrics. ‘x’ indicates the according feature was included in training and evaluating for the experiment.

Exp. ID	Patient Information	Code Description	Code Names	Past Episode	Data Sampling	Specificity	Sensitivity	F1 Score	AUROC Score
1	x					0.968 (0.037)	0.066 (0.086)	0.088 (0.096)	0.628 (0.034)
2	x	x				**0.972 (0.02)**	0.1 (0.053)	0.161 (0.075)	0.658 (0.014)
3	x		x			0.971 (0.021)	0.097 (0.051)	0.15 (0.073)	0.655 (0.013)
4	x		x	x		0.954 (0.037)	0.148 (0.088)	0.2 (0.097)	0.686 (0.02)
5	x		x	x	x	0.55 (0.069)	**0.726 (0.066)**	**0.369 (0.0098)**	**0.7 (0.01)**

## Data Availability

Data are unavailable due to privacy or ethical restrictions.

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
