# Peer review of "Forecasting Patient Early Readmission from Irish Hospital Discharge Records Using Conventional Machine Learning Models"

_diagnostics, 2024, doi:10.3390/diagnostics14212405_

Round 1

Reviewer 1 Report

Comments and Suggestions for Authors

Reviewer Comments
The objective of this paper is to forecast patient readmissions based on electronic health records (EHR) from Irish acute hospitals. The authors utilized conventional machine learning techniques such as logistic regression, random forest, AdaBoost, and XGBoost, with a focus on predicting patient readmissions within 30 days. Additionally, they applied explainable AI to analyze the correlation between diagnosis codes and readmissions.

Major Comments

1.     The authors did not provide a clear logical or statistical explanation of the relationship between the risk factors and patient readmission.

2.     The protocol for patient selection is not explained, leaving ambiguity around how patients were chosen for the study.

3.     Given the highly imbalanced nature of the data, the authors have not explained how they mitigated bias in the results. Furthermore, they did not provide any statistical evidence to support their approach to handling imbalance.

4.     The authors did not provide a complete overview of the dataset attributes. Including a sample dataset would improve transparency and allow for better evaluation of the study.

5.     A power analysis was not conducted, making it unclear whether the sample size used is sufficient to ensure the robustness of the results.

6.     Advanced techniques, such as deep learning, have already been applied to similar studies, particularly for predicting readmissions in cardiovascular patients and infants. The authors should address these approaches.

7.     The authors could strengthen their study by incorporating comparisons with the following works:

a.      Yu, Maggie, Mark Harrison, and Nick Bansback. "Can prediction models for hospital readmission be improved by incorporating patient-reported outcome measures? A systematic review and narrative synthesis." Quality of Life Research (2024): 1-13.

b.     Yu, Min-Young, and Youn-Jung Son. "Machine learning–based 30-day readmission prediction models for patients with heart failure: a systematic review." European Journal of Cardiovascular Nursing (2024): zvae031.

c.      Zarghani, Abolfazl. "Comparative Analysis of LSTM Neural Networks and Traditional Machine Learning Models for Predicting Diabetes Patient Readmission." arXiv preprint arXiv:2406.19980 (2024).

8.     A more comprehensive and detailed analysis of the results is necessary to demonstrate the consistency and reliability of the findings.

9.     The accuracy (0.701) and AUROC (0.006) are somewhat moderate and could be improved further by utilizing techniques like bagging or hybrid machine learning models.

10.  The study lacks proper benchmarking against existing models or datasets, which is essential for assessing the performance of their approach.

11.  The authors did not sufficiently explain the rationale behind using correlation analysis or how it influenced the final outcomes of the study.

12.  The representation of AUC curves is missing, which is a critical component in evaluating the performance of classification models.

13.  Conclusion: The study would greatly benefit from improvements in accuracy, a more thorough data analysis, and a deeper explanation of the methodologies used.

Comments on the Quality of English Language

The authors' English writing is clear and well-structured, making the paper easy to follow

Reviewer 2 Report

Comments and Suggestions for Authors

Forecasting patient early readmission from Irish hospital discharge records using conventional machine  learning models

Review

A brief summary

The paper focuses on predicting patient readmission to improve healthcare outcomes and reduce costs. Researchers compared various machine learning models using electronic discharge records from an Irish hospital. They found that XGBoost, which uses patient demographics, hospitalization history, and clinical diagnoses, performed the best in predicting readmissions within 30 days. The study also used SHAP (SHapley Additive exPlanations) values to identify which diagnoses codes were significant predictors of patient readmission. This helps healthcare professionals understand and potentially mitigate the risks associated with certain conditions.

Overall, the study highlights the effectiveness of using routinely collected hospital data and explainable AI techniques for predicting patient readmissions.

General concept comments

Article:

The article is well-written, but it needs more clarification. Specifically, it should avoid using initials without providing details, and it should further explain the features of the model, as these are a crucial part of the article's motivations.

Review:

v  The number of citations is small. Additional citations should be added.

o   "Predicting Hospital Readmission via Cost-sensitive Deep Learning" lai

o   "Thirty-day hospital readmission prediction model based on common data model with weather and air quality data". Scientific reports. Nature Portfolio. 2021. Borim Ryu et al.

v  In paragraph 3.2.1 , perhaps SMOTE technic can be also used for undersampling. It considered a good method.

v  Table 4. Page 7 lines 272-273. F1  score should be additionally presented. It is important to verify that XGBoost has high F1-score specifically in cases of imbalances datasets.

v Figure 4 on page 10 presents feature importance; however, the precise meaning of each feature is not clear. The features should be explained in more detail.

Specific comments:

v  In the abstract line 6 SHAP is mentioned in initials should be written in details: SHAP (SHapley Additive exPlanations).

Comments on the Quality of English Language

No comments

Author Response

Please see the attachment. Under the section "Reviewer#2"
